# Emergence and Spread of B.1.1.7 Lineage in Primary Care and Clinical Impact in the Morbi-Mortality among Hospitalized Patients in Madrid, Spain

**DOI:** 10.3390/microorganisms9071517

**Published:** 2021-07-15

**Authors:** Laura Martínez-García, Marco Antonio Espinel, Melanie Abreu, José María González-Alba, Desirèe Gijón, Amaranta McGee, Rafael Cantón, Juan Carlos Galán, Jesús Aranaz

**Affiliations:** 1Servicio de Microbiología, Hospital Universitario Ramón y Cajal, 28034 Madrid, Spain; laura.martinez.garcia@salud.madrid.org (L.M.-G.); melanie.abreu@salud.madrid.org (M.A.); jmgonzalba@yahoo.es (J.M.G.-A.); desiree.gijon@salud.madrid.org (D.G.); rafael.canton@salud.madrid.org (R.C.); 2Instituto Ramón y Cajal de Investigación Sanitaria (IRYCIS), 28034 Madrid, Spain; marcoantonio.espinel@salud.madrid.org (M.A.E.); jesusmaria.aranaz@salud.madrid.org (J.A.); 3Centro de Investigación Biomédica en Red (CIBER) en Epidemiología y Salud Pública, 28029 Madrid, Spain; 4Servicio de Medicina Preventiva, Hospital Universitario Ramón y Cajal, 28034 Madrid, Spain; 5Red Española de Investigación en Patología Infecciosa (REIPI), 28009 Madrid, Spain; 6Facultad de Ciencias de la Salud, Universidad Internacional de la Rioja, 26006 Logroño, Spain

**Keywords:** SARS-CoV-2, B1.1.7 lineage, variants, replacement, burden of disease

## Abstract

In December 2020, UK authorities warned of the rapid spread of a new SARS-CoV-2 variant, belonging to the B.1.1.7 lineage, known as the Alpha variant. This variant is characterized by 17 mutations and 3 deletions. The deletion 69–70 in the spike protein can be detected by commercial platforms, allowing its real-time spread to be known. From the last days of December 2020 and over 4 months, all respiratory samples with a positive result for SARS-CoV-2 from patients treated in primary care and the emergency department were screened to detect this variant based on the strategy S gene target failure (SGTF). The first cases were detected during week 53 (2020) and reached >90% of all cases during weeks 15–16 (2021). During this period, the B.1.1.7/SGTF variant spread at a rapid and constant replacement rate of around 30–36%. The probability of intensive care unit admission was twice higher among patients infected by the B.1.1.7/SGTF variant, but there were no differences in death rate. During the peak of the third pandemic wave, this variant was not the most prevalent, and it became dominant when this wave was declining. Our results confirm that the B.1.1.7/SGTF variant displaced other SARS-CoV-2 variants in our healthcare area in 4 months. This displacement has led to an increase in the burden of disease.

## 1. Introduction

In the last days of December 2019, the first cases of a pneumonia of unknown origin were described, and only 1 month later, the World Health Organization (WHO) declared a public health emergency of international concern. A pandemic caused by a new coronavirus known as severe acute respiratory syndrome coronavirus 2 (SARS-CoV-2) started, causing great disruption on a global scale [1,2]. SARS-CoV-2 virus, closely related to other lethal coronaviruses such as MERS-CoV (Middle East respiratory syndrome) and SARS-CoV-1, has shown a higher capacity of transmission than its known relatives. Currently, near 190 million people have been diagnosed worldwide (https://coronavirus.jhu.edu/map.html, accessed on 14 July 2021). Each replication event is a biological opportunity for mutation, and each transmission event is an opportunity for the spreading of these emerged mutants. Consequently, the diversification of SARS-CoV-2 variants has grown continuously, evolving towards a complex description of lineages (https://www.gisaid.org, accessed on 14 July 2021). During the pandemic period, few variants have required attention. They are known as variants of concern (VOCs), and their surveillance is required by international organizations such as the CDC and ECDC [3]. These variants have been responsible for different epidemiological waves in many countries. Among them, the variant carrying the D614G mutation in the spike (S) protein, associated with the B.1 lineage, quickly replaced the original strains from China responsible for the first pandemic wave [4]. Several works have demonstrated that variants carrying D614G mutation yield a higher replication rate and higher viral load than the wild-type variant [5]. In early summer, a new variant emerged in Spain belonging to the B.1.177 lineage, characterized by a new mutation A222V added to D614G in protein S. This variant was associated with the second wave in multiple European countries [6]. Currently, up to five VOCs are being monitored, but this number might increase in the coming months. SARS-CoV-2 VOCs were initially named using the name of the country of origin; however, the WHO has recently recommended using the Greek alphabet for these variants (e.g., Alpha variant (B.1.1.7/501Y.V1), Beta variant (B.1.351/501Y.v2) and Gamma variant (P1/501Y.V3)).

The B.1.1.7 lineage emerged in September 2020 in South East England (Kent county, UK), although the England Public Health authorities did not announce a 3.7-fold increase in the COVID-19 cases associated with this new variant until December [7]. Shortly after, it was described at least in 83 countries [8]. Recently, the transmission rate was estimated as 43–90%, clearly higher than the pre-existing variant [9] with a similar impact to that observed in other countries such as the United States [10]. Although the impact of the virulence is not clear, there is controversy about a higher pathogenicity, especially among patients under 70 years old [11]. This variant is characterized by 17 nonsynonymous mutations and 3 deletions, including N501Y mutation associated with higher transmissibility [9], P681H mutation related to the enhancement of virus entry into cells and 69–70 deletion in protein S. In Spain, the B.1.1.7 lineage was first detected in the last days of 2020 [12], and for 4 months we systematically performed screening to detect this variant among all SARS-CoV-2-positive samples analyzed by any platform used in our laboratory. The aim of this study was to evaluate how the new variant has replaced the other ones circulating in our geographic area, to study its transmissibility and to understand the associated burden of disease.

## 2. Materials and Methods

### 2.1. Clinical Samples

The study was carried out in Hospital Universitario Ramon y Cajal for 4 months (2nd January 2021–30th April 2021). In this period, all respiratory samples (nasopharyngeal swabs, sputum and bronchoalveolar lavage fluid) that yielded a positive result for SARS-CoV-2 and cycle threshold (Ct) value ≤30 were tested to screen for B.1.1.7 lineage screening based on the strategy S gene target failure (SGTF) using multiplex TaqPath COVID-19 assay (Thermo Fisher, Waltham, MA, United States). The del 69–70 in S gene is present in multiple lineages but was used for rapid screening of B.1.1.7/SGTF because of its strong correlation. At the end of February, we additionally included a double screening based on the single nucleotide polymorphism (SNP) N501Y. Only one sample per patient was included. The samples were received from 20 primary care (PC) settings and the emergency department, representing around 10% of all population living in Madrid. Those patients admitted to the emergency department were divided into patients requiring hospitalization and those who could be discharged. Moreover, demographic data (date of birth, gender) were obtained for all patients; epidemiological follow-up data of hospitalized patients were collected, such as clinical data (date of admission in the intensive care unit (ICU), date of admission in the intermediate respiratory care unit (IRCU), reason for follow-up interruption).

The study was approved by the ethical committee from our center (reference number 099/21).

### 2.2. Sequencing Analysis and Phylogenetic Analysis

A total of 100 samples in which loss of amplification in S gene target was observed were sequenced by whole-genome sequencing by next-generation sequencing (NGS) to confirm the previous assignation to the B.1.1.7 lineage based on screening protocol. These samples were selected during the analyzed period. Sequencing was performed following the Artic protocol v3 (https://artic.network/ncov-2019, accessed on 14 July 2021) [13]. First, we performed RNA extraction of nasopharyngeal swab samples by Microlab Nimbus (Hamilton, Bonaduz AG, Switzerland). Preparation of cDNA and multiplex PCR were performed. Clean-up and size selection were performed by Agencourt AMPure XP (Beckman Coulter, Brea, CA, United States). For library preparations, we followed the Illumina DNA Prep protocol. Libraries were fully sequenced in an Illumina MiSeq instrument (Illumina, San Diego, CA, United States). The raw data generated in binary base call (BCL) format from MiSeq was demultiplexed to FASTQ files using bcl2fastq v2.20. The raw reads were assembled by mapping to the reference genome from Wuhan, China (hCoV-19/Wuhan/Hu-1/2019, GenBank accession number: NC_045512.2), using Illumina DRAGEN COVID Lineage v2.3.2–v3.5.3, which also generates a consensus sequence. The parameters used were those chosen by default by the software itself. The consensus sequence obtained was uploaded to Pangolin COVID-19 Lineage Assigner (https://pangolin.cog-UK.io/, accessed on 14 July 2021) [14], using several versions during the course of the study, from Pangolin 2.0 to Pangolin v3.1 [15]. All these programs are available at https://emea.illumina.com/informatics/biological-interpretation/coronavirus-software.html (accessed on 14 July 2021). Moreover, the sequences obtained belonging to the B.1.1.7 lineage, as well as B.1.1.7 SARS-CoV-2 genomes from England available on the GISAID database, were aligned using MAFFT program v7.477 (https://mafft.cbrc.jp/alignment/software/, accessed on 14 July 2021) and then manually revised using MEGA X program (https://www.megasoftware.net/, accessed on 14 July 2021) to correct misaligned sequences as a consequence of artefactual frameshifts. The phylogenetic tree was reconstructed by maximum likelihood method (ML) with FastTree using GTR +I + G nucleotide substitution model. Bootstrap values were estimated using the SH test (support > 95%).

### 2.3. Statistical analysis

The descriptive analysis of quantitative variables included the median and interquartile range (IQR). For the categorical variables, the odds ratio (OR), the percentage and the 95% confidence interval (95% CI) are provided. Comparisons between medians were performed using Mann–Whitney U test, and comparisons between categorical variables were compared using the χ2 test and the Fisher exact test. Statistical significance was set at *p* < 0.05. Univariate and multivariate analyses (logistic regression) were performed to study factors associated with mortality. The multivariate analyses included variables that had reached statistical significance in the univariate analysis and those thought to be relevant although they did not reach statistical significance. These analyses were repeated according to two groups of age (patients ≤65 and >65 years).

## 3. Results

### 3.1. Replacement Rate of B.1.1.7/SGTF (Alpha Variant) in Outpatients in Our Healthcare Area

During the studied period, 27,633 respiratory samples coming from 20 PC centers and the emergency department of Hospital Universitario Ramón y Cajal were processed for diagnosing COVID-19 infection: 20,870 samples (75.5%) were received from primary care and 6763 (24.5%) samples were received from patients admitted to the emergency department. In the case of the primary care-based samples, 3920 (18.8%) yielded a positive result for SARS-CoV-2, whereas 1555 patients from the emergency department were infected by SARS-CoV-2 (22.9%). The screening started in week 52 of 2020, and only four cases were detected in week 53 in two PC centers (representing 1.3%), but they were not included in this analysis. Following the temporal evolution in primary care, in the first 2 weeks of the study period, 15 out of 574 positive samples (2.6%) were initially classified as B.1.1.7/SGTF, coming from 8/20 primary care centers (Figure 1a). In the third and fourth weeks, the prevalence of this new variant reached 10.5% (123 suspected cases/1174 screened samples), and it spread to 18/20 centers (Figure 1b)

At the end of the first month, this variant was present in practically all health areas. In the successive two biweekly periods, the B.1.1.7/SGTF was suspected in all centers, and the prevalence continued growing from 27.3 to 46.3% in February (Figure 1c–d). It is important to highlight as in that moment, the proportion of the B.1.1.7/SGTF was higher than 75% in five PC centers. In the next 2 months, this variant reached 75.1% and 92.7% in March and April respectively (Figure 1e–h), although in the last 2 weeks the replacement reached a plateau. According to these data, the variant belonging to B.1.1.7/SGTF showed a linear increase (R > 0.99) with a viral replacement rate of around 30% during the described period. The shift speed was practically constant along the studied period. A similar replacement was observed in the emergency department during the study period, growing from 4.6 to 91.3% from the first to the eighth biweekly period (Figure 2). The viral replacement rate was similar in the emergency department (R > 0.99), although the shift replacement was slightly higher between 2 and 3 months (reaching a replacement speed around 36%). To confirm our initial classification, 38 samples yielding positive amplification for SARS-CoV-2 and an amplification pattern suggestive of B.1.1.7 lineage were sequenced. All of them were correctly assigned to the B.1.1.7 lineage, and accession numbers are available in Appendix A.

### 3.2. Burden of Disease in Hospitalized Patients

During this period, 1555 patients with confirmed SARS-CoV-2 infection were admitted to the hospital. Among them, 426 (37.7%) were suspected of infection by B.1.1.7 lineage over the study period. Median age of patients with B.1.1.7/SGTF was 68 years old, while the median age of patients with non-B.1.1.7 variant was 71 years old (*p* = 0.004) (Table 1). This difference was more marked when we analyzed the population group >65 years (78 vs. 82, *p* = 0.001) (Appendix A). Slightly more than 19.5% of patients with B.1.1.7/SGTF required admission in the ICU, while only 10.3% of patients with other variants needed intensive care (*p* = 0.001) (Table 1). The probability of ICU admission was twice higher among patients with the B.1.1.7 lineage (OR 2.11 95 CI% = 1.55 − 2.87). The UCI admission risk was 3 times higher in patients aged older than 65 carrying B.1.1.7/SGTF. Similar findings were observed among the patients admitted to IRCU (8.7% vs. 4.4% for B.1.1.7 lineage vs. non-B.1.1.7 lineage) (*p* = 0.001), with patients aged older than 65 carrying B.1.1.7/SGTF having 2-fold higher probability of IRCU admission (OR 2.05; 95% CI = 1.32 − 3.19). However, death rate was slightly lower in patients infected with the studied variant (13.9% vs. 15.6%), but this difference was not significant. Therefore, there was no association between the probability of death and the presence of B.1.1.7/SGTF (Table 1).

## 4. Discussion

The fact that the Alpha variant (B.1.1.7) lineage can serendipitously be detected by ThermoFisher TaqPath COVID-19 PCR assay has allowed exhaustive surveillance of this variant, thus helping to understand its spreading dynamics in the population. The molecular detection of SARS-CoV-2 in our laboratory was performed using different platforms; however, all positive samples detected in any of them were reanalyzed using the ThermoFisher assay. In week 53, 2020, we detected the first four positive cases (which are not included in the sampling) in patients from primary care settings. Over the next 4-month period, a practically complete replacement was observed among all SARS-CoV-2 variants. During this period, the B.1.1.7/SGTF lineage reached >90% of all SARS-CoV-2 variants circulating in Madrid, but it showed a plateau in the last 2 weeks. The replacement speed was similar in both primary care and the emergency department (around 30–36%), slightly lower than previous data, which suggested an increased transmission of 35–90%, according to mathematical models [9,11]. Curiously, this replacement was not accompanied by a drastic increase in the number of cases. Madrid experienced a third pandemic wave between mid-December and the end of February, where the accumulated incidence reached 993 and 559 cases/100,000 population in January and February, respectively (www.mscbs.gob.es, accessed on 14 July 2021) [16,17], whereas the incidence of B.1.1.7/SGTF kept growing in these months from 14.3% in primary care and 11.7% in the emergency department at the end of January to 44.9% and 37.5%, respectively, at the end of February. In the last days of April, the accumulated incidence decreased to 335, whereas the percentage of B.1.1.7/SGTF reached >90%. Therefore, confirming previous results, we also observed a higher transmission rate for B.1.1.7/SGTF, but we did not find that this replacement was coincident with a new pandemic wave.

On the other hand, the B.1.1.7/SGTF variant was also initially associated with more severe illness [18]. In our study, we found no differences in the mortality rate among groups diagnosed with the B.1.1.7/SGTF compared to other variants. These results are concordant with other published studies [19,20]. On the other hand, our results suggest that patients infected by the B.1.1.7/SGTF have a 2 times higher risk of admission into the ICU (reaching 3 times in patients aged >65 years) and the IRCU than patients infected by non-B.1.1.7 lineage. These results are different from those reported in a recently published study [21] in which the authors did not find differences in severity between patients grouped by lineage. However, in this study, the follow-up was only 14 days after the onset of symptoms. In our research, the study period was since hospital diagnosis, which might imply a longer follow-up period. However, our results are similar to those reported in a community-based study [22], as well as in a recent European study, which showed a 2.3 times higher risk of being admitted to ICU for people infected by B.1.1.7, compared to non-VOC cases [23]. Currently, the most recent variants, such as the Beta (B.1.351/501Y.v2) and Gamma (P1/501Y.V3) variants, represent few cases, and we do not know as these variants could modify the molecular epidemiology in the next months.

Our study has some limitations. First, data about the clinical characteristics, comorbidities and vaccination of our patients were not available; therefore, we were not able to explore the influence of vaccination on the transmission and the burden of disease. However, attending to Spanish vaccination strategy, in the months of our study, the population >80 years old started the vaccination program; therefore, we cannot exclude that the finding of high admission to the ICU in the >65 years group could be related to the non-vaccinated status of those <80 years old. A second limitation was the impact of the rapid antigen detection test (RADT). In Spain, the use of this test represents 40% of SARS-CoV-2 detection. This value reaches 60% in primary care in our study. These samples were not sent to the reference laboratory; therefore, we could not know the SARS-CoV-2 variant in these cases.

In conclusion, our results show that B.1.1.7/SGTF is also associated with increased hospitalization and more severe infection [24]. Patients infected with this variant seem to have a higher need for ICU and IRCU admission. Finally, more prospective studies are needed in order to evaluate the influence of patients’ comorbidities in the B.1.1.7/SGTF infection and the sequelae of the disease in recovered COVID patients.

## Figures and Tables

**Figure 1 microorganisms-09-01517-f001:**
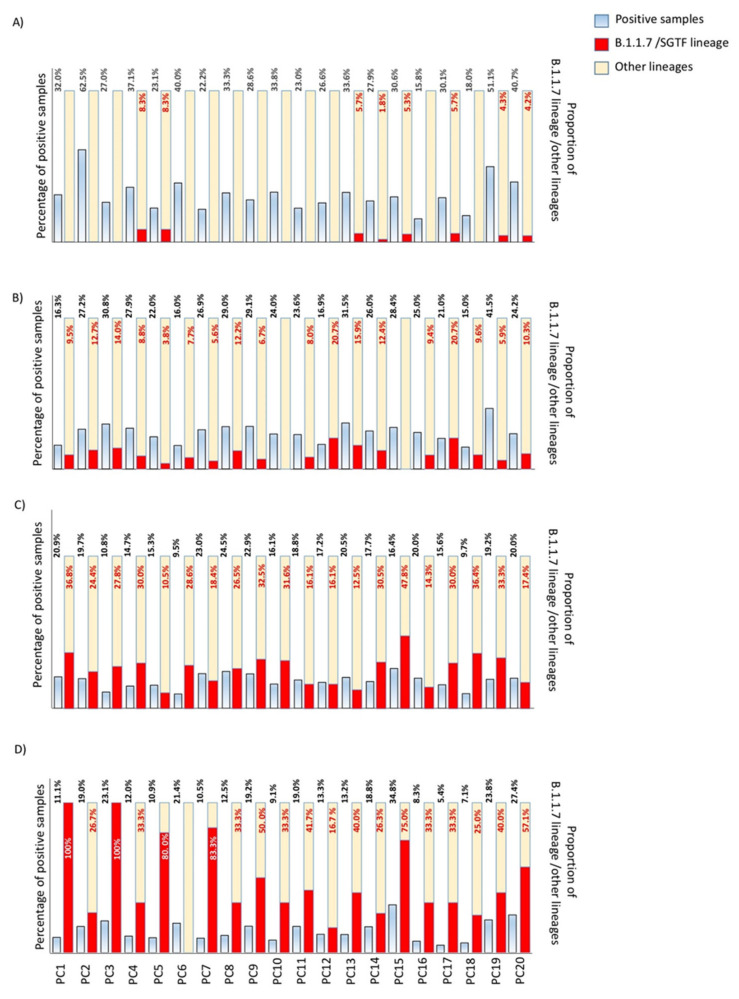
Temporal evolution in the incidence of B.1.1.7 (Alpha variant) in our health area: (**A**) first biweekly period (2–17 January); (**B**) second biweekly period (18–31 January); (**C**) third biweekly period (1–14 February); (**D**) fourth biweekly period (15–28 February); (**E**) fifth biweekly period (1–14 March); (**F**) sixth biweekly period (15–28 March); (**G**) seventh biweekly period (29 March–11 April); (**H**) eighth biweekly period (12–22 April). All dates correspond to 2021. PC: primary care.

**Figure 2 microorganisms-09-01517-f002:**
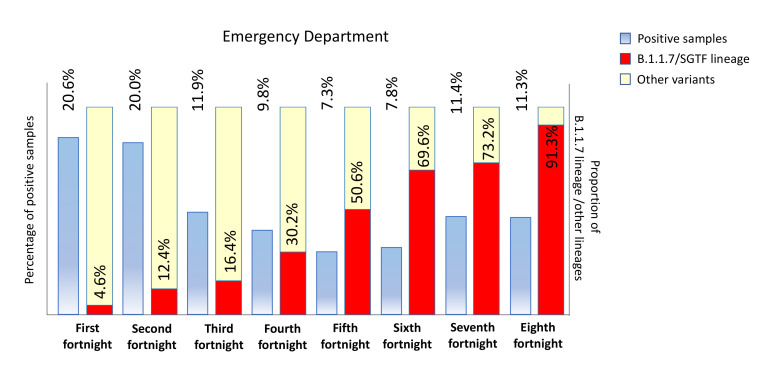
Temporal evolution in the incidence of B.1.1.7 (Alpha variant) in the emergency department from 02 January to 22 April 2021. SGTF: strategy S gene target failure.

**Table 1 microorganisms-09-01517-t001:** Demographic and clinical characteristics of the patients infected by B.1.1.7 and non-B.1.1.7 lineages. IQR: interquartile range, OR: odds ratio, 95%CI: 95% confidence interval, ICU: intensive care unit, IRCU: intensive respiratory care unit, sig: significance.

Variable	Total of Patients			
	B.1.1.7/SGTF (n = 426)	No B.1.1.7 (n = 1129)	*p*	OR
Age (median)	68	71	0.004	
IQR	(56–79)	(57–83)		
Sex				
Women	41.1%	44.3%	0.25	
Men	58.9%	55.7%		
ICU admission	19.5%	10.3%	0.001	2.11
(95%CI)	(15.83–23.57)	(8.56–12.19)		(1.55–2.87)
IRCU admission	8.7%	4.4%	0.001	2.05
(95%CI)	(6.19–11.77)	(3.3–5.79)		(1.32–3.19)
Death	13.9%	15.6%	0.49	0.87
(95%CI)	(10.52–17.95)	(13.47–17.87)		(0.62–1.23)

## Data Availability

The sequences were deposited in GISAID (https://www.gisaid.org, accessed on 14 July 2021). Accession numbers are shown in Appendix A.

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
