# Peer review of "Emergence and Spread of B.1.1.7 Lineage in Primary Care and Clinical Impact in the Morbi-Mortality among Hospitalized Patients in Madrid, Spain"

_microorganisms, 2021, doi:10.3390/microorganisms9071517_

Round 1
Reviewer 1 Report
The results are clearly presented, and explained.
Nevertheless, the B.1.1.7 variant subject is less concerning today than 6 months earlier.
I would like the authors to explain why they did not use more specific screening methods than the one based on SGTF ?
I would like to know if the authors could know the vaccination rate in both groups ?
Thank you for the answers.
Best regards.
Author Response
Reviewer 1
Comments and Suggestions for Authors
1.1. The results are clearly presented, and explained.
Thank you for the comment.
1.2 Nevertheless, the B.1.1.7 variant subject is less concerning today than 6 months earlier.
Yes, it is true. Currently, the international concern is the spreading of SARS-CoV-2 delta variants. Of course, we are also monitoring the fast (almost explosive) increase in our health care area. We want to share with you the current scenario. (Please, see the attachment).
A month after our study was submitted, Alpha variant (following the last nomenclature recommended by WHO), B.1.1.7 in our manuscript, was dominant in our population. In that moment, the impact of Delta variant was irrelevant. This epidemiological situation did not change until mid-June (weeks 23-24) and we sent the manuscript in week 22. We are examining the fast dynamics of viral replacement, and in order to have a current and fully formed picture, we agree that more manuscripts will be necessary to better address viral evolution, especially with new variants arising. It is exciting to understand the viral ecological phenomenon, which needs to continue with future research.
This manuscript was designed, approved by the Ethical Committee, analyzed and written in a short amount of time.
1.3 I would like the authors to explain why they did not use more specific screening methods than the one based on SGTF?
We only used the S-deletion in the first cases, because it was the only available strategy to suspect the presence of alpha variant. For this reason, in the first days, many of them were sequenced. Moreover, from an epidemiological point of view, the presence of other SARS-CoV-2 variants carrying S-deletion, such as B.1.525 (Eta variant in WHO nomenclature), are exceptionally described in Madrid (and so far in Spain) during the study period. According to official data from the reference laboratory of our country (Instituto de Salud Carlos III) only 10 sequences were identified in Spain belonging to eta variant. Based on our data, 383 complete sequences were obtained from January to May. Among them 275/383 were alpha variant (B.1.1.7) and only three belonged to eta variant (B.1.525).
On the other hand, since the month of February, we included the strategy based on N501Y detection, using the Thermo Fisher proposal, because the B.1.258 (carrying also the del 69-70) was increasing in some regions of Spain. The N501Y change allowed us to avoid a potential misclassification of the studied variants. However, we did not detect the profile SGTF (+) and N501Y (-) in the period studied.
Therefore, we though that the impact of other non-alpha variants in the observed shift was residual. We have included a comment in the main text to address this potential situation.
1.4 I would like to know if the authors could know the vaccination rate in both groups?
The vaccination rate in Spain was lower than the planned targets during the first months. According to official data (Ministry of Health, in www.epdata.es) only 7% of people living in Madrid had been vaccinated at the end of April. Moreover, the vaccination started with nursing home patients and health care personnel. In fact, in Madrid, at the end of April, only 471,507 people had been completely vaccinated, with full doses and 1,447,286 with the first dose. (https://www.mscbs.gob.es/profesionales/saludPublica/ccayes/alertasActual/nCov/pbiVacunacion.htm).
Adressing the reviewer’s comment we have accessed the vaccination status of our sample subjects. Only 14 patients were fully vaccinated with 2 doses (Pfizer-BioNTech, 9 in the non-B.1.1.7 group and 5 in B.1.1.7 group). It is a sample too small to analyze in order to actually find any statistical differences.
On the other hand, during this period we detected several cases of COVID-19 infection among fully vaccinated healthcare personnel. Moreover, an outbreak among institutionalized elder people with two doses of Pfizer vaccine was detected in June, being B.1.1.7 lineage, the SARS-CoV-2 variant responsible. All patients (health personnel and institutionalized elder people) had good evolution.

Reviewer 2 Report
The author presented an interesting overview of SARS-CoV-2 variant evolution within the Madrid area. They need to improve the "material and methods" providing further details in order to make the study reproducible.
Mainly in the sequencing and statistical analysis paragraph there is a lack of information:
- which sequencing platform was used for NGS?
- how did you obtain the consensus sequences?
- which modification were made in the sequences during the manual curation using MEGA?
- you must specify which variables did you use for multivariate analyses and which of them are statistical significant or not (you can put these data in supplementary materials)
Explicit in the legend of figure 1 the abreviation of "PC"
Round 2
Reviewer 2 Report
I would like to thank the authors for the replies and the efforts done.
In my opinion the manuscript is not yet sufficiently improved.
The authors must provide the name and version of any software used for the genome assembly (or a reference if the analysis pipeline was already published).
In the comments they said "PC is the abbreviation of Primary Care. We introduce this information in text of figure 1" but this information is not present.
accordingly to the "instructions for authors":
- They should be described with sufficient detail to allow others to replicate and build on published results. New methods and protocols should be described in detail while well-established methods can be briefly described and appropriately cited. Give the name and version of any software used and make clear whether computer code used is available. Include any pre-registration codes.
- Acronyms/Abbreviations/Initialisms should be defined the first time they appear in each of three sections: the abstract; the main text; the first figure or table. When defined for the first time, the acronym/abbreviation/initialism should be added in parentheses after the written-out form.
Author Response
Dear Reviewer 2;
We would like to try to reply to your questions.
The authors must provide the name and version of any software used for the genome assembly (or a reference if the analysis pipeline was already published).
When the sequences are obtained, encrypted data flow from MiSeq into BaseSpace Sequence Hub for analysis data. For assembled, mapping, and consensus sequence we used Illumina® DRAGEN COVID Lineage but obviously different versions from v2.3.2- v3.5.3 depending on the updates. We always used the parameters chosen by default by the software itself. This program has the possibility to upload the consensus sequences in Pangolin COVID-19 Lineage Assigner (https://pangolin.cog-UK.io/). This possibility was used to allocate the corresponding sequences in a lineage. One more time, in this short time, we used different Pangolin versions (from 2.0 to 3.1). Up to this point, this process is commonly used in many laboratories working in sequencing. You can obtain these programs at https://emea.illumina.com/informatics/biological-interpretation/coronavirus-software.html
Now in the new version, we wrote:
First, we performed RNA extraction of nasopharyngeal swab samples by Microlab STARlet (Hamilton). Preparation of cDNA and multiplex PCR was performed. Clean-up and size selection was performed by Agencourt AMPure XP (Beckman Coulter). For library preparations, we followed the Illumina DNA Prep protocol. Libraries were fully sequenced in an Illumina MiSeq instrument (Illumina®). The raw data generated in binary base call (BCL) format from MiSeq was demultiplexed to FASTQ files using bcl2fastq v2.20. The raw reads were assembled by mapping to the reference genome from Wuhan, China (hCoV-19/Wuhan/Hu-1/2019, GenBank accession number: NC_045512.2), using Illumina® DRAGEN COVID Lineage v2.3.2- v3.5.3, which also generates a consensus sequence. The parameters used were those chosen by default by the software itself. The consensus sequence obtained was upload to Pangolin COVID-19 Lineage Assigner (https://pangolin.cog-UK.io/) [14] using, during this time several versions, from Pangolin 2.0 and to Pangolin v3.1 [15]. All these programs are available at https://emea.illumina.com/informatics/biological-interpretation/coronavirus-software.html. Moreover, the sequences obtained belonging to B.1.1.7 lineage, as well as B.1.1.7 SARS-CoV-2 genomes from England available on the GISAID database were aligned using MAFFT program v7.477 (https://mafft.cbrc.jp/alignment/software/) and then manually revised using MEGA X program (https://www.megasoftware.net/) to correct misaligned sequences as a consequence of artefactual frameshifts. The Phylogenetic tree was reconstructed by Maximum Likelihood Method (ML) with FastTree using GTR +I+G nucleotide substitution model. Bootstrap values were estimated using the SH test (support > 95%).
I know that the nomenclature for SARS-CoV-2 is a dynamic process; Sometimes the sequences assigned to determined lineage could be changed using different Pangolin versions or when the number of sequences related to the proposal lineage increases. This possibility has not been described in B.1.1.7 lineage; however, we also wanted to re-analyze the sequences using phylogenetic reconstructions to confirm the results obtained from DRAGEN COVID.
In the comments they said "PC is the abbreviation of Primary Care. We introduce this information in the text of figure 1" but this information is not present
Sorry, it was a human error. It is modified
accordingly to the "instructions for authors":
- They should be described with sufficient detail to allow others to replicate and build on published results. New methods and protocols should be described in detail while well-established methods can be briefly described and appropriately cited. Give the name and version of any software used and make clear whether computer code used is available. Include any pre-registration codes.
it is possible that we did not explain well to the reviewer, but we did not use new methods or protocols. We did not develop a new pipeline. As you commented in our first reply, we used the most common pipelines distributed by Illumina. Now, we have included in the main text those well-known names and versions used for the assignation of lineages, which are public access on the webpage.
- Acronyms/Abbreviations/Initialisms should be defined the first time they appear in each of three sections: the abstract; the main text; the first figure or table. When defined for the first time, the acronym/abbreviation/initialism should be added in parentheses after the written-out form.
We have revised the acronyms in the abstract, main text, figures, and table
We have modified in abstract ICU for intensive care unit.
In the main text, we include the complete name of MERS-CoV-2 as Middle East respiratory syndrome or Ct as Cycle threshold (Ct)
In the main text next-generation sequencing (NGS)
OR odds ratio in material and methods
In figure 1, PC: Primary care
In figure 2, SGTF: strategy S gene target failure
We consider that other acronyms such as CDC or ECDC are well-known. Moreover, the names of programs are known according to their abbreviation, we did not consider include for instance: MAFFT as Multiple Alignment using Fast Fourier Transform. MEGA as Molecular Evolutionary Genetics Analysis
